# CBLUE: A Chinese Biomedical Language Understanding Evaluation Benchmark

**Mosha Chen** *   **Zhen Bi** *   **Xiaozhuan Liang** *   **Lei Li** *   **Ningyu Zhang** †   **Xin Shang**

**Kangping Yin**   **Chuanqi Tan**   **Jian Xu**   **Fei Huang**   **Luo Si**   **Yuan Ni**   **Guotong Xie**

**Zhifang Sui**   **Baobao Chang**   **Hui Zong**   **Zheng Yuan**   **Linfeng Li**   **Jun Yan**

**Hongying Zan**   **Kunli Zhang**   **Buzhou Tang**†   **Qingcai Chen** †

CBLUE Team ‡

## Abstract

Artificial Intelligence (AI), along with the recent progress in biomedical language understanding, is gradually offering great promise for medical practice. With the development of biomedical language understanding benchmarks, AI applications are widely used in the medical field. However, most benchmarks are limited to English, which makes it challenging to replicate many of the successes in English for other languages. To facilitate research in this direction, we collect real-world biomedical data and present the first Chinese Biomedical Language Understanding Evaluation (CBLUE) benchmark: a collection of natural language understanding tasks including named entity recognition, information extraction, clinical diagnosis normalization, single-sentence/sentence-pair classification, and an associated online platform for model evaluation, comparison, and analysis. To establish evaluation on these tasks, we report empirical results with the current 11 pre-trained Chinese models, and experimental results show that state-of-the-art neural models perform by far worse than the human ceiling. Our benchmark is released at `https://tianchi.aliyun.com/dataset/dataDetail?dataId=95414&lang=en-us`.

## 1   Introduction

Artificial intelligence is gradually changing the landscape of healthcare, and biomedical research [35]. With the fast advancement of biomedical datasets, biomedical natural language processing (BioNLP) has facilitated a broad range of applications such as biomedical text mining, which leverages textual data in Electronic Health Records (EHRs). For example, BioNLP methods can be employed to provide recommendations for specialized healthcare to those most at risk during pandemics (COVID-19) using the text and information in EHRs.

A key driving force behind such improvements and rapid iterations of models is the use of general evaluation datasets and benchmarks [9]. Pioneer benchmarks, such as BLURB [10], PubMedQA

---

*Equal contribution and shared co-first authorship.

†Corresponding author.

‡Author contributions are listed in the appendix.

[13], and others, have provided us with the opportunity to conduct research on biomedical language understanding and developing real-world applications. Unfortunately, most of these benchmarks are developed in English, which makes the development of the associated machine intelligence Anglo-centric. Meanwhile, other languages, such as Chinese, have unique linguistic characteristics and categories that need to be considered. Even though Chinese speakers account for a quarter of the world population, there have been no existing Chinese biomedical language understanding evaluation benchmarks.

To address this issue and facilitate natural language processing studies in Chinese, we take the first step in introducing a comprehensive **C**hinese **B**iomedical **L**anguage **U**nderstanding **E**valuation (**CBLUE**) benchmark with eight biomedical language understanding tasks. These tasks include named entity recognition, information extraction, clinical diagnosis normalization, short text classification, question answering (in transfer learning setting), intent classification, semantic similarity, and so on.

We evaluate several pre-trained Chinese language models on CBLUE and report their performance. The current models still perform by far worse than the standard of single-human performance, leaving room for future improvements. We also conduct a comprehensive analysis using case studies to indicate the challenges and linguistic differences in Chinese biomedical language understanding. We intend to develop a universal GLUE-like open platform for the Chinese BioNLP community, and this work is a small step in that direction. Overall, the main contributions of this study are as follows:

- We propose the first Chinese biomedical language understanding benchmark, an open-ended, community-driven project with eight diverse tasks. The proposed benchmark serves as a platform for the Chinese BioNLP community and encourages new dataset contributions.

- We report a systematic evaluation of 11 Chinese pre-trained language models to understand the challenges derived by these tasks. We release the source code of the baselines as a toolkit at `https://github.com/CBLUEbenchmark/CBLUE` for future research purposes.

## 2 Related Work

Several benchmarks have been developed to evaluate general language understanding over the past few years. GLUE [29] is one of the first frameworks developed as a formal challenge affording straightforward comparison between task-agnostic transfer learning techniques. SuperGLUE [28], styled after GLUE, introduce a new set of more difficult language understanding tasks, a software toolkit, and a public leaderboard. Other similarly motivated benchmarks include DecaNLP [22], which recast a set of target tasks into a general question-answering format and prohibit task-specific parameters, and SentEval [2], which evaluate explicitly fixed-size sentence embeddings. Non-English benchmarks include RussianSuperGLUE [25] and CLUE [34], which is a community-driven benchmark with nine Chinese natural language understanding tasks. These benchmarks in the general domain provide a north star goal for researchers and are part of the reason we can confidently say we have made great strides in our field.

For BioNLP, many datasets and benchmarks have been proposed [30, 18, 33] which promote the biomedical language understanding [1, 17, 16]. Tsatsaronis et al. [27] propose biomedical language understanding datasets as well as a competition on large-scale biomedical semantic indexing and question answering. Jin et al. [13] propose PubMedQA, a novel biomedical question answering dataset collected from PubMed abstracts. Pappas et al. [23] propose BioRead, which is a publicly available cloze-style biomedical machine reading comprehension (MRC) dataset. Gu et al. [10] create a leaderboard featuring the Biomedical Language Understanding & Reasoning Benchmark (BLURB). Unlike a general domain corpus, the annotation of a biomedical corpus needs expert intervention and is labor-intensive and time-consuming. Moreover, most of the benchmarks are based on English; ignoring other languages means that potentially valuable information may be lost, which can be helpful for generalization.

In this study, we focus on Chinese and aim to develop the first Chinese biomedical language understanding benchmark. Note that Chinese is linguistically different from English and other Indo-European languages, necessitating an evaluation BioNLP benchmark designed explicitly for Chinese.

| Dataset | Task | Train | Dev | Test | Metrics |
|---------|------|-------|-----|------|---------|
| CMeEE | NER | 15,000 | 5,000 | 3,000 | Micro F1 |
| CMeIE | Information Extraction | 14,339 | 3,585 | 4,482 | Micro F1 |
| CHIP-CDN | Diagnosis Normalization | 6,000 | 2,000 | 10,192 | Micro F1 |
| CHIP-STS | Sentence Similarity | 16,000 | 4,000 | 10,000 | Macro F1 |
| CHIP-CTC | Sentence Classification | 22,962 | 7,682 | 10,000 | Macro F1 |
| KUAKE-QIC | Intent Classification | 6,931 | 1,955 | 1,994 | Accuracy |
| KUAKE-QTR | Query-Document Relevance | 24,174 | 2,913 | 5,465 | Accuracy |
| KUAKE-QQR | Query-Query Relevance | 15,000 | 1,600 | 1,596 | Accuracy |

Table 1: Task descriptions and statistics in CBLUE. CMeEE and CMeIE are sequence labeling tasks. Others are single sentence or sentence pair classification tasks.

# 3 CBLUE Overview

CBLUE consists of 8 biomedical language understanding tasks in Chinese. We will introduce the task definitions, detailed data collection procedures, and characteristics of CBLUE followingly.

## 3.1 Tasks

**CMeEE** Chinese Medical Named Entity Recognition, a dataset first released in CHIP2020[4], is used for CMeEE task. Given a pre-defined schema, the task is to identify and extract entities from the given sentence and classify them into nine categories: disease, clinical manifestations, drugs, medical equipment, medical procedures, body, medical examinations, microorganisms, and department.

**CMeIE** Chinese Medical Information Extraction, a dataset that is also released in CHIP2020 [11], is used for CMeIE task. The task is aimed at identifying both entities and relations in a sentence following the schema constraints. There are 53 relations defined in the dataset, including 10 synonymous sub-relationships and 43 other sub-relationships.

**CHIP-CDN** CHIP Clinical Diagnosis Normalization, a dataset that aims to standardize the terms from the final diagnoses of Chinese electronic medical records, is used for the CHIP-CDN task. Given the original phrase, the task is required to normalize it to standard terminology based on the International Classification of Diseases (ICD-10) standard for Beijing Clinical Edition v601.

**CHIP-CTC** CHIP Clinical Trial Classification, a dataset aimed at classifying clinical trials eligibility criteria, which are fundamental guidelines of clinical trials defined to identify whether a subject meets a clinical trial or not [38], is used for the CHIP-CTC task. All text data are collected from the website of the Chinese Clinical Trial Registry (ChiCTR) [5], and a total of 44 categories are defined. The task is like text classification; although it is not a new task, studies and corpus for the Chinese clinical trial criterion are *still limited*, and we hope to promote future researches for social benefits.

**CHIP-STS** CHIP Semantic Textual Similarity, a dataset for sentence similarity in the non-i.i.d. (non-independent and identically distributed) setting, is used for the CHIP-STS task. Specifically, the task aims to transfer learning between disease types on Chinese disease questions and answer data. Given question pairs related to 5 different diseases (The disease types in the training and testing set are different), the task intends to determine whether the semantics of the two sentences are similar.

**KUAKE-QIC** KUAKE Query Intent Classification, a dataset for intent classification, is used for the KUAKE-QIC task. Given the queries of search engines, the task requires to classify each of them into one of 11 medical intent categories defined in KUAKE-QIC, including diagnosis, etiology analysis, treatment plan, medical advice, test result analysis, disease description, consequence prediction, precautions, intended effects, treatment fees, and others.

---

[4]http://cips-chip.org.cn/
[5]http://chictr.org.cn/

**KUAKE-QTR**   KUAKE Query Title Relevance, a dataset used to estimate the relevance of the title of a query document, is used for the KUAKE-QTR task. Given a query (e.g., "Symptoms of vitamin B deficiency"), the task aims to find the relevant title (e.g., "The main manifestations of vitamin B deficiency").

**KUAKE-QQR**   KUAKE Query-Query Relevance, a dataset used to evaluate the relevance of the content expressed in two queries, is used for the KUAKE-QQR task. Similar to KUAKE-QTR, the task aims to estimate query-query relevance, which is an essential and challenging task in real-world search engines.

## 3.2   Data Collection

Since machine learning models are mostly data-driven, data plays a critical role, and it is pretty often in the form of a static dataset [8]. We collect data for different tasks from diverse sources, including clinical trials, EHRs, medical books, and search logs from real-world search engines. As biomedical data may contain private information such as the patient's name, age, and gender, **all collected datasets are anonymized and reviewed by the IRB committee of each data provider to preserve privacy.** We introduce the data collection details followingly.

**Collection from Clinical Trials**

Clinical trial eligibility criteria text is collected from ChiCTR, a non-profit organization that provides registration for clinical trial information for public research use. Eligibility criteria text is organized as a paragraph in the inclusion criteria and exclusion criteria in each trial registry file. Meaningless text was excluded, such as "The criteria is as follows", and the remained text was annotated to generate the CHIP-CTC dataset.

**Collection from EHRs**

We obtain the final diagnoses of the medical records from several Class A tertiary hospitals and sample a few diagnosis items from different medical departments to construct the CHIP-CDN dataset for research purposes. The diagnosis items are randomly sampled from the items not covered by the common medical synonyms dict. **No privacy information is involved in the final diagnoses.**

**Collection from Medical Forum and Textbooks**

Due to the COVID-19 pandemic, online consultation becomes more and more popular via the Internet. To promote data diversity, we select the online questions by patients to build the CHIP-STS dataset. Note that most of the questions are chief complaints. To ensure the authority and practicability of the corpus, we also select medical textbooks of Pediatrics [31], Clinical Pediatrics [26] and Clinical Practice[6]. We collect data from these sources to construct the CMeIE and CMeEE datasets.

**Collection from Search Engine Logs**

We also collect search logs from real-world search engines like the Alibaba KUAKE Search Engine[7]. First, we filter the search queries in the raw search logs by the medical tag to obtain candidate medical texts. Then, we sample the documents for each query with non-zero relevance scores (i.e., to determine if the document is relevant to the query). Specifically, we divide all the documents into three categories, namely high, middle, and tail documents, and then uniformly sample the data to guarantee diversity. We leverage the data from search logs to construct KUAKE-QTC, KUAKE-QTR, and KUAKE-QQR datasets.

## 3.3   Annotation

Each sample is annotated by three to five crowd workers, and the annotation with the majority of votes is taken to estimate human performance. During the annotation phase, we add control questions

---

[6]http://www.nhc.gov.cn/
[7]https://www.myquark.cn/

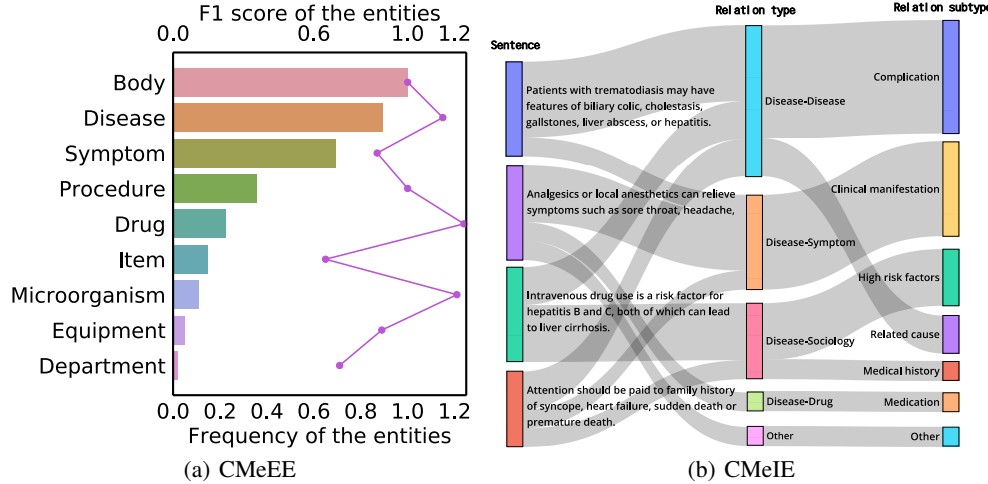

(a) CMeEE

(b) CMeIE

Figure 1: Analysis of the named entity recognition and information extraction datasets. (a) illustrates the entity (coarse-grained) distribution in CMeEE and the impact of data distribution on the model's performance. We set entity type Body with the maximum number of entities to 1.0, and others to the ratio of number or F1 score to Body. (b) shows the relation hierarchy in CMeIE.

to prevent dishonest behaviors by the crowd workers. Consequently, we reject any annotations made by crowd workers who fail in the training phase and do not adopt the results of those who achieved low performance on the control tasks. We maintain strict and high criteria for approval and review at least 10 random samples from each worker to decide whether to approve or reject all their HITs. We also calculate the average inter-rater agreement between annotators using Fleiss' Kappa scores [7], finding that five out of six annotations show almost perfect agreement ($\kappa = 0.9$).

## 3.4 Characteristics

**Utility-preserving Anonymization**   Biomedical data may be considered as a breach in the privacy of individuals because they usually contain sensitive information. Thus, we conduct utility-preserving anonymization following [15] to anonymize the data before releasing the benchmark.

**Real-world Distribution**   To promote the generalization of models, all the data in our CBLUE benchmark follow real-world distribution without up/downsampling. As shown in Figure 1(a), our dataset follows long-tail distribution following Zipf's law so that all data will inevitably be long-tailed. However, long-tail distribution has no significant effect on performance. Further, some datasets, such as CMedIE, have label hierarchy with both coarse-grained and fine-grained relation labels, as shown in Figure 1(b).

**Diverse Tasks Setting**   Our CBLUE benchmark includes eight diverse tasks, including named entity recognition, relation extraction, and single-sentence/sentence-pair classification. Besides the independent and i.i.d. scenarios, our CBLUE benchmark also contains a specific transfer learning scenario supported by the CHIP-STS dataset, in which the testing set has a different distribution from the training set.

## 3.5 Leaderboard

We provide a leaderboard for users to submit their own results on CBLUE. The evaluation system will give final scores for each task when users submit their prediction results. The platform offers 60 free GPU hours from Aliyun[8] to help researchers develop and train their models.

---

[8]https://tianchi.aliyun.com/notebook-ai/

### 3.6 Distribution and Maintenance

Our CBLUE benchmark was released online on April 1, 2021. Up to now, more than **300** researchers have applied the dataset, and over **80** teams have submitted their model predictions to our platform, including medical institutions (Peking Union Medical College Hospital, etc.), universities (Tsinghua University, Zhejiang University, etc.), and companies (Baidu, JD, etc.). We will continue to maintain the benchmark by attending to meet new requests and adding new tasks.

| Model | CMeEE | CMeIE | CDN | CTC | STS | QIC | QTR | QQR | Avg. |
|---|---|---|---|---|---|---|---|---|---|
| BERT-base | 62.1 | 54.0 | 55.4 | 69.2 | 83.0 | 84.3 | 60.0 | **84.7** | 69.1 |
| BERT-wwm-ext-base | 61.7 | 54.0 | 55.4 | 70.1 | 83.9 | 84.5 | 60.9 | 84.4 | 69.4 |
| RoBERTa-large | 62.1 | 54.4 | 56.5 | **70.9** | 84.7 | 84.2 | 60.9 | 82.9 | 69.6 |
| RoBERTa-wwm-ext-base | 62.4 | 53.7 | 56.4 | 69.4 | 83.7 | 85.5 | 60.3 | 82.7 | 69.3 |
| RoBERTa-wwm-ext-large | 61.8 | **55.9** | 55.7 | 69.0 | 85.2 | 85.3 | 62.8 | 84.4 | 70.0 |
| ALBERT-tiny | 50.5 | 35.9 | 50.2 | 61.0 | 79.7 | 75.8 | 55.5 | 79.8 | 61.1 |
| ALBERT-xxlarge | 61.8 | 47.6 | 37.5 | 66.9 | 84.8 | **84.8** | 62.2 | 83.1 | 66.1 |
| ZEN | 61.0 | 50.1 | 57.8 | 68.6 | 83.5 | 83.2 | 60.3 | 83.0 | 68.4 |
| MacBERT-base | 60.7 | 53.2 | 57.7 | 67.7 | 84.4 | 84.9 | 59.7 | 84.0 | 69.0 |
| MacBERT-large | **62.4** | 51.6 | **59.3** | 68.6 | **85.6** | 82.7 | **62.9** | 83.5 | 69.6 |
| PCL-MedBERT | 60.6 | 49.1 | 55.8 | 67.8 | 83.8 | 84.3 | 59.3 | 82.5 | 67.9 |
| Human | 67.0 | 66.0 | 65.0 | 78.0 | 93.0 | 88.0 | 71.0 | 89.0 | 77.1 |

Table 2: Performance of baseline models on CBLUE benchmark.

### 3.7 Reproducibility

To make it easier to use the CBLUE benchmark, we also offer a toolkit implemented in PyTorch [24] for reproducibility. Our toolkit supports mainstream pre-training models and a wide range of target tasks. Different from existing pre-training model toolkits [37], the toolkit is aimed at fast validating performance on the CBLUE benchmark.

## 4 Experiments

**Baselines** We conduct experiments with baselines based on different Chinese pre-trained language models. We add an additional output layer (e.g., MLP) for each CBLUE task and fine-tune the pre-trained models. Code for reproducibility is available in `https://github.com/CBLUEbenchmark/CBLUE`.

**Models** We evaluate CBLUE on the following public available Chinese pre-trained models:

- BERT-base [5]. We use the base model with 12 layers, 768 hidden layers, 12 heads, and 110 million parameters.

- BERT-wwm-ext-base [4]. A Chinese pre-trained BERT model with whole word masking.

- RoBERTa-large [21]. Compared with BERT, RoBERTa removes the next sentence prediction objective and dynamically changes the masking pattern applied to the training data.

- RoBERTa-wwm-ext-base/large. RoBERTa-wwm-ext is an efficient pre-trained model which integrates the advantages of RoBERTa and BERT-wwm.

- ALBERT-tiny/xxlarge [14]. ALBERT is a pre-trained model with two objectives: Masked Language Modeling (MLM) and Sentence Ordering Prediction (SOP), which shares weights across different layers in the transformer.

- ZEN [6]. A BERT-based Chinese text encoder enhanced by N-gram representations, where different combinations of characters are considered during training.

- Mac-BERT-base/large [3]. Mac-BERT is an improved BERT with novel MLM as a correction pre-training task, which mitigates the discrepancy of pre-training and fine-tuning.

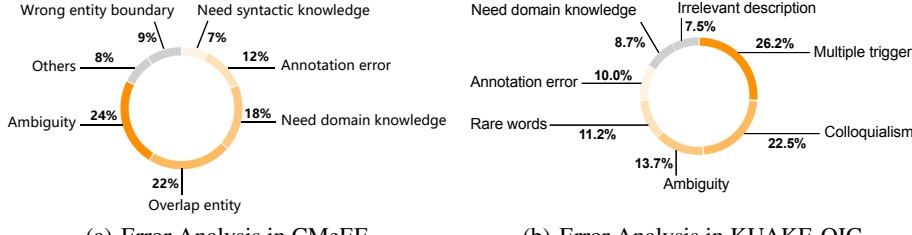

|                | (a) Error Analysis in CMeEE | (b) Error Analysis in KUAKE-QIC |
|---|---|---|

Figure 2: We conduct error analysis on dataset CMeEE and QIC. For CMeEE, we divide error cases into 6 categories, including ambiguity, need domain knowledge, overlap entity, wrong entity boundary, annotation error, and others (long sequence, rare words, etc.). For KUAKE-QIC, we divide error cases into 7 categories, including multiple triggers, colloquialism, ambiguity, rare words, annotation error, irrelevant description, and need domain knowledge.

- PCL-MedBERT[9]. A pre-trained medical language model proposed by the Intelligent Medical Research Group at the Peng Cheng Laboratory, with excellent performance in medical question matching and named entity recognition.

|  |  | CMeEE | CMeIE | CDN | CTC | STS | QIC | QTR | QQR |
|---|---|---|---|---|---|---|---|---|---|
| **Trained annotation** | annotator 1 | 69.0 | 62.0 | 60.0 | 73.0 | 94.0 | 87.0 | 75.0 | 80.0 |
|  | annotator 2 | 62.0 | 65.0 | 69.0 | 75.0 | 93.0 | 91.0 | 62.0 | 88.0 |
|  | annotator 3 | 69.0 | 67.0 | 62.0 | 80.0 | 88.0 | 83.0 | 71.0 | 90.0 |
|  | avg | 66.7 | 64.7 | 63.7 | 76.0 | 91.7 | 87.0 | 69.3 | 86.0 |
|  | majority | **67.0** | **66.0** | **65.0** | **78.0** | **93.0** | **88.0** | **71.0** | **89.0** |
|  | best model | 62.4 | 55.9 | 59.3 | 70.9 | 85.6 | 85.5 | 62.9 | 84.7 |

Table 3: Human performance of two-stage evaluation scores with the best-performed model. "avg" refers to the mean score from the three annotators. "majority" indicates the performance taken from the majority vote of amateur humans. Bold text denotes the best result among human and model prediction.

We implement all baselines with PyTorch [24]. Note that BERT-base, ALBERT-tiny/xxlarge, and RoBERTa-large are representatives of pre-trained language models. BERT-wwm-ext-base, RoBERTa-wwm-ext-base/large, ZEN, Mac-BERT-base/large utilize the specific characteristics (e.g., words and phrases) of the Chinese language. PCL-MedBERT further utilize domain-adaptive pre-training [12], which can consistently improve performance on tasks in the biomedical domain. We tune all the hyper-parameters based on the performance of each model on the development set. We implement each experiment five times and calculate the average performance. All the training details can be found in the appendix.

## 4.1 Benchmark Results

We report the results of our baseline models on the CBLUE benchmark in Table 2. We notice that the model obtain better performance with larger pre-trained language models. We also observe that models which use whole word masking do not always yield better performance than others in some tasks, such as CTC, QIC, QTR, and QQR, indicating that tasks in our benchmark are challenging, and more sophisticated technologies should be developed. Further, we find that ALBERT-tiny achieves comparable performance to base models in tasks of CDN, STS, QTR, and QQR, illustrating that smaller models may also be efficient in specific tasks. Finally, we notice that PCL-MedBERT, which tends to be state-of-the-art in Chinese biomedical text processing tasks, while does not perform as well as we expected. This further demonstrates the difficulty of our benchmark, and contemporary models may find it difficult to quickly achieve outstanding performance.

---

[9]https://code.ihub.org.cn/projects/1775

## 4.2 Human Performance

For all of the tasks in CBLUE, we ask human amateur annotators with no medical experience to label instances from the testing set and compute the annotators' majority vote against the gold label annotated by specialists. Similar to SuperGLUE [28], we first need to train the annotators before they work on the testing data. Annotators are asked to annotate some data from the development set; then, their annotations are validated against the gold standard. Annotators need to correct their annotation mistakes repeatedly so that they can master the specific tasks. Finally, they annotate instances from the testing data, and these annotations are used to compute the final human scores. The results are shown in Table 3 and the last row of Table 2. In most tasks, humans tend to behave better than machine learning models. We analyze the human performance detailedly in the next section.

| Sentence | Word | Label | RO | MB |
|---|---|---|---|---|
| 血液生化分析的结果显示维生素B缺乏率约为12%～19%。 | 血液生化分析 | Ite | Pro | Pro |
| The results of blood biochemical analysis show that vitamin B lack rate is about 12% to 19%. | blood biochemical analysis | Ite | Pro | Pro |
| 皮疹可因宿主产生特异性的抗毒素抗体而减少。 | 抗毒素抗体 | Bod | O | Bod |
| The rash can be reduced by the host producing specific anti-toxin antibodies. | anti-toxin antibodies | Bod | O | Bod |
| 根据遗传物质的结构和功能改变的不同，可将遗传病分为五类：1.染色体病指染色体数目异常，或者染色体结构异常，包括缺失、易位、倒位等 | 缺失, 易位, 倒位 | Sym, Sym, Sym | O | Sym, Sym, Sym |
| According to the structure and function of genetic material, genetic diseases are divided into five categories: 1. Chromosomal diseases refer to abnormal chromosome number or chromosome structure abnormalities, including deletions, translocations, inversions... | deletions, translocations, inversions | Sym, Sym, Sym | O | Sym, Sym, Sym |

Table 4: Case studies in CMeEE. We evaluate roberta-wwm-ext and PCL-MedBERT on 3 sampled sentences, with their gold labels and model predictions. Ite (medical examination items), Pro (medical procedure), Bod (body), and Sym (clinical symptoms) are labeled for medical named words. O means that the model fails to extract the entity from sentences. RO=roberta-wwm-ext, MB=PCL-MedBERT.

## 4.3 Case studies

We choose two datasets: CMeEE and KUAKE-QIC, a sequence labeling and classification task, respectively, to conduct case studies. As shown in Figure 2, we report the statistics of the proportion of various types of error cases[10]. For CMeEE, we notice that *overlap entity*, *ambiguity*, *need domain knowledge*, *annotation error* are major reasons that result in the prediction failure. Furthermore, there exist many instances with *overlap entity*, which may lead to confusion for the named entity recognition task. While in the analysis for KUAKE-QIC, almost half of bad cases are due to *multiple triggers* and *colloquialism*. *Colloquialism* is natural in search queries, which means that some descriptions of the Chinese medical text are too simplified, colloquial, or inaccurate.

We show some cases on CMeEE in Table 4. In the second row, we notice that given the instance of "皮疹可因宿主产生特异性的抗毒素抗体而减少 (*Rash can be reduced by the host producing specificanti-toxin antibodies.*)", ROBERTA and PCL-MedBERT obtain different predictions. The reason is that there exist medical terminologies such as "抗毒素抗体 (*anti-toxin antibodies*)". ROBERTA can not identify those tokens correctly, but PCL-MedBERT, pre-trained on the medical corpus, can successfully make it. Moreover, PCL-MedBERT can accurately extract entities "缺失,易位,倒位 (*eletions, translocations, inversions*)" from the long sentences, which is challenging for other models.

---

[10]See definitions of errors in the appendix.

We further show some cases on KUAKE-QIC in Table 5. In the first case, we notice that both BERT and BERT-ext fail to obtain the intent label of the query "请问淋巴细胞比率偏高、中性细胞比率偏低有事吗? (*Does it matter if the ratio of lymphocytes is high and the ratio of neutrophils is low?*)", while MedBERT can obtain the correct prediction. Since "淋巴细胞比率 (*ratio of lymphocytes*)" and "中性细胞比率 (*ratio of neutrophils*)" are biomedical terminologies, and the general pre-trained language model has to leverage domain knowledge to understand those phrases. Moreover, we observe that all model obtain incorrect predictions for the query "咨询：请问小孩一般什么时候出水痘 (*Consultation: When do children usually get chickenpox?*)" in the second case. Note that there exists lots of colloquial text in search queries (*colloquialism*), which have different distributions, thus, mislead the model predictions.

| Query | Model | | | Gold |
| --- | --- | --- | --- | --- |
| | BERT | BERT-ext | MedBERT | |
| 请问淋巴细胞比率偏高、中性细胞比率偏低有事吗? 
 Does it matter if the ratio of lymphocytes is high and the ratio of neutrophils is low? | 病情诊断 

 Diagnosis | 病情诊断 

 Diagnosis | 指标解读 

 Test results analysis | 指标解读 

 Test results analysis |
| 咨询：请问小孩一般什么时候出水痘? 
 Consultation: When do children usually get chickenpox? | 其他 
 Other | 其他 
 Other | 其他 
 Other | 疾病表述 
 Disease description |
| 老人收缩压160，舒张压只有40多，是什么原因？怎么治疗? 
 The systolic blood pressure of the elderly is 160, and the diastolic blood pressure is only more than 40. What is the reason? How to treat? | 病情诊断 

 Diagnosis | 病情诊断 

 Diagnosis | 病情诊断 

 Diagnosis | 治疗方案 

 Treatment |

Table 5: Case studies in KUAKE-QIC. We evaluate the performance of baselines with 3 sampled instances. The correlation between Query and Title is divided into 3 levels (0-2), which means '*poorly related or unrelated*', '*related*' and '*strongly related*'. BERT = BERT-base, BERT-ext = BERT-wwm-ext-base, MedBERT = PCL-MedBERT.

As shown in Table 4 and Table 5, compared with other languages, Chinese language is very colloquial even in medical texts. Furthermore, Chinese is also a tonal language, and the meaning of a word changes according to its tone, which usually causes confusion and difficulties for machine reading. In summary, we conclude that **tasks in CBLUE are not easy to solve since the Chinese language has unique characteristics**, and more robust models that fully understand the semantics of Chinese, especially the informal or formal usages in the medical domain, should be taken into consideration.

## 4.4 Limitations

Although our CBLUE offers diverse settings, there are still some tasks not covered by the benchmark, such as medical dialogue generation [20, 19, 36] or medical diagnosis [32]. We encourage researchers in both academics and industry to contribute new datasets. Besides, our benchmark is static; thus, models may still achieve outstanding performance on tasks but fail on simple challenge examples and falter in real-world scenarios. We leave this as future works to construct a platform including dataset creation, model development, and assessment, leading to more robust and informative benchmarks.

## 4.5 Conclusion and Future Work

In this paper, we present a Chinese Biomedical Language Understanding Evaluation (CBLUE) benchmark, which consists of eight natural language understanding tasks, along with an online leaderboard for model evaluation. We evaluate 11 current language representation models on CBLUE and analyzed their results. The results illustrate the limited ability of state-of-the-art models to handle some of the more challenging tasks. In contrast to English benchmarks such as GLUE/SuperGLUE and BLURB, whose model performance already matches human performance, we observe that this is far from the truth for Chinese biomedical language understanding. We hope our benchmark can help promote developing stronger natural language understanding models in the future.

## 4.6 Broader Impact

The COVID-19 (coronavirus disease 2019) pandemic has had a significant impact on society, both because of the severe health effects of COVID-19 and the public health measures implemented to slow its spread. A lack of information fundamentally causes many difficulties experienced during the outbreak; attempts to address these needs caused an information overload for both researchers and the public. Biomedical natural language processing—the branch of artificial intelligence that interprets human language—can be applied to address many of the information needs making urgent by the COVID-19 pandemic. Unfortunately, most language benchmarks are in English, and no biomedical benchmark currently exists in Chinese. Our benchmark CBLUE, as the first Chinese biomedical language understanding benchmark, can serve as an open testbed for model evaluations to promote the advancement of this technology.

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
