# OpenReview forum: "CBLUE: A Chinese Biomedical Language Understanding Evaluation Benchmark"
_NeurIPS.cc/2021/Track/Datasets_and_Benchmarks/Round1 — Submitted to NeurIPS 2021 Datasets and Benchmarks Track (Round 1)_

### Official Review · Reviewer_Qu9S · 2021-07-02
**A GLUE-like benchmark for Chinese Biomedical NLP**

**Rating:** 5
**Confidence:** 3
**Correctness:** Yes

**Strengths:**

* A platform for evaluating Chinese Biomedical NLP models, convering various tasks.
* A bunch of existing BERT-based models are evaluated on the proposed benchmark.

**Weaknesses:**

* The novely and target audience of proposed benchmark seems limited, because it follows previous efforts (GLUE, CLUE) and builds a domain-specific language-specific benchmark.
* The authors mention that the data are proprietary (no license?), and this may pose restrictions on researchers to use it.


**Additional Feedback:**

* Section 1, 'Anglo-centric' may not be the proper word, since most of these datasets you mention are US-based.
* Section 3.3, Fleiss' Kappa seems unbelievable high. 0.9 is not 'moderate', it may be described as 'almost perfect agreement'

**Clarity:**

* The paper is well written.

**Documentation:**

* Yes

**Relation To Prior Work:**

* Yes

**Summary And Contributions:**

This paper presents a benchmark for Chinese Biomedical NLP.
It contains eight datasets (a combination of existing and new datasets), including NER, classification, measuring sentence similarity.
These tasks are framed as either sentence-level or token-level tasks.

In addition to scholarly articles, clinical notes, this benchmark contains queries of search engines, which to my knowledge is under explored data source.

---

> ### Author Response · Authors · 2021-07-14
> **Response to Reviewer3**
>
> Thanks for your constructive feedback. We will carefully revise the paper.
>
> Our CBLUE is a domain-specific language-specific benchmark, but it still has some special characters.
>
> 1.All the data in our CBLUE benchmark follow real-world distribution without up/downsampling. As shown in Figure 1, our dataset follows long-tail distribution following Zipf’s law so that all data will inevitably be long-tailed.
>
> 2.Besides the independent and i.i.d. scenarios, our CBLUE benchmark also contains a specific transfer learning scenario supported by the CHIP-STS dataset, in which the testing set has a different distribution from the training set.
>
> 3.To make it easier to use the CBLUE benchmark, we also offer a toolkit implemented in PyTorch for reproducibility.
>
> Sorry for some missing parts. We release the datasets and benchmark following the CC BY-NC 4.0 license.

---

### Official Review · Reviewer_gPg8 · 2021-07-02

**Rating:** 6
**Confidence:** 3

**Strengths:**

- The authors unified eight biomedical language understanding tasks in Chinese to form a benchmark, which bio-data is relatively sensitive and hard to obtain.
- The dataset has been applied and investigated by researchers in the field.

**Weaknesses:**

- No dataset comparison with existing Chinese understanding tasks, for example, CLUE (https://github.com/CLUEbenchmark/CLUE), in terms of size, task diversity, etc.
- I would expect to have some results and baselines that are fine-tuned from existing Chinese NLU work. For example, what is the gap between standard Chinese NLU and Chinese bio-NLU?

**Additional Feedback:**

N/A

**Clarity:**

- Please bold the highest performance in Table 2.
- Figure 1 (a) is not informative, please provide details of each ratio.

**Correctness:**

- My biggest concern about the correctness is that whether the collected annotation is representative and correct.
- For example, for a well-trained annotator (based on the training description in section 3.3), the F1 scores in several tasks such as CMeEE, CMeIE, and CDN are only 60-70% as human performance.
- Therefore, I have a concern that whether the annotation could be different if we ask experts and doctors to do the job?
- What is the confident from 1 to 10 that the authors believe their gold labels are correct?

**Documentation:**

No significant error is found.

**Relation To Prior Work:**

As I mentioned in the weakness part, I believe it is better to compare with the existing Chinese NLU datasets that are not necessary to be in the bio-domain.

**Summary And Contributions:**

This paper proposes a new benchmark called CBLUE, Chinese Biomedical Language Understanding Evaluation, for medical language understanding.

---

> ### Author Response · Authors · 2021-07-14
> **Response to Reviewer2**
>
> Thanks for your constructive feedback. We will carefully revise the paper.
>
> The most significant difference from existing Chinese understanding benchmarks such as CLUE is the task diversity and data distribution. Our CBLUE includes some tasks such as query-document relevance, diagnosis normalization, which is not involved in CLUE. Moreover, the CBLUE contains a non-i.i.d setting task in CHIP-CTC.
>
> We will add more experimental results to illustrate the gap between standard Chinese NLU and Chinese bio-NLU. We think the domain gap is a major difference. Pre-trained language model in the general domain may fail in such bio-NLU tasks. Furthermore, we notice that PCL-MedBERT, which tends to be state-of-the-art in Chinese biomedical text processing tasks, does not perform as well as we expected. This demonstrates the difficulty of our benchmark, and contemporary models may find it difficult to quickly achieve outstanding performance. For both the data collection and human baseline, we use Fleiss' Kappa to evaluate the agreement among annotators (The scores indicate that annotators have almost perfect agreement).
>
> In the data collection stage, we ask experts and doctors (not amateur crowd workers) to annotate all the corpus. Each annotator is paid from 0.5 to 1 RMB per instance according to the task complexity. We maintain strict and high criteria for approval and review at least 10 random samples from each worker to decide whether to approve or reject.
>
> In the human evaluation in section 4.3, we ask the amateur workers to evaluate human performance. All workers are trained before annotating instances from the testing data.

---

### Official Review · Reviewer_zcSC · 2021-07-02
**Amazing resource of epic scope with a few suggested documentation and writing improvements.**

**Rating:** 6
**Confidence:** 4

**Strengths:**

This paper describes a vast amount of work to both introduce datasets and a benchmark that uses them and make the benchmark accessible by providing introductory notebooks that work on servers with free GPU hours. In addition to the expansive starter code, there is also a lot of exhaustive technical documentation of the dataset formats, which makes working with the data much easier.

**Weaknesses:**

My main concern with this paper is about its lack of documentation of anything that is not the data format. There is insufficient information about the crowdsourcing process and no consideration of negative impacts, both of which are crucial in the biomedical domain where high-quality annotations are incredibly important.

Specifically, my points of critique are

1. The benchmark, while including much long-tailed data, does not explicitly differentiate between those examples and head examples in the reporting of accuracy, which can lead to an overestimate of real-world utility. Summarizing performance with a single accuracy number has been rightfully critiqued and new benchmarks should not fall into the same trap.
2. It reads from the description in 4.2 that all the crowd workers were untrained. While I sincerely hope that this was not the case for the collection of the data, this requires a disclaimer in the text itself that the “human performance” is not the performance of a trained professional, but that of amateurs with no medical experience. For example, in the table it should be labeled as “majority vote of amateur humans”. The section also doesn’t mention how many humans were chosen to annotate each datapoint and how many dev data points they had to annotate during training, or how long the task took or how much was paid. Moreover, for both the data collection and human baseline, there was no provided information about annotator agreement or other data quality indicators.
3. Nowhere in 4.3 does it state why the case studies are particular to the Chinese language, instead of general findings that you would also see in other languages (note that I do not want to discredit the value of non-English datasets, just that the particular instances don’t say much about Chinese)


**Additional Feedback:**

While the KUAKE datasets sound super interesting, I wonder if it is actually too different from the other tasks and may be worth more exploring in a separate benchmark.


**Clarity:**

There are a couple of unclear points in the text:

- How was the “meaningless text” found in the CHIP-CTC processing?
- While it is stated that no PII is included in the data, it would be nice to empirically verify this or describe the process through which this is ensured, similar to how MIMIC does it
- How were diagnosis items sampled for CHIP-CDN


**Correctness:**

AI is not changing medical practice, as suggested by the abstract, and instead any system should be critically examined (see, e.g., https://www.nature.com/articles/s42256-021-00307-0)


**Documentation:**

In addition to the weaknesses in documentation pointed out above, there are couple minor issues:

- For CMeIE and CMeEE, what are the licenses of the underlying textbooks that you can use them?
- How much were crowd workers paid, what was their qualification, how long did they take for each annotation task
- Question 4d, 5a/b/c are all answered with yes, but I couldn’t find this in any of the writeup or the supplementary pdf
- The data cards for each dataset are purely descriptive and do not describe the annotation or curation process.


**Ethics:**

There are two potential concerns with this work:

1. The data uses amateurs to annotate biomedical data, which can lead to mislabeled examples and a misrepresentation of a model's performance. If a model was chosen based on numbers on the benchmark, this could cause real-world harm.
2. It lowers the bar of entry to work with biomedical data. While generally a good thing, it may dilute the pool of data-driven work in the biomedical field even more than it already it, making it hard for experts to spot the relevant work.

The broader impact section does not discuss either of the two concerns and neither of them are mentioned at all. While in Question 1c of the checklist, the authors refer to supplementary material regarding negative social impact, I could not find this.


**Relation To Prior Work:**

n/a

**Summary And Contributions:**

This paper introduces CBLUE, a chinese biomedical NLU benchmark. As part of assembling the benchmark, the authors annotated multiple datasets for different tasks that cover a wide spectrum of biomedical tasks. The authors provide many baseline numbers and an (amateur) human performance estimate, alongside in-depth documentation on how to use the data.

Overall, if the documentation issues could be addressed and the evaluation process expanded, I'd be okay with seeing this paper be accepted.

---

> ### Author Response · Authors · 2021-07-14
> **Response to  Reviewer1**
>
> Thanks for your constructive feedback. We will carefully revise the paper and add more introduction about the crowdsourcing process and its negative impacts.
>
> 1.We have carefully revised figure 1 and add the performance of each relation with the long-tailed distribution. We observe that the distribution has an impact on the model performance. We will try to utilize more metrics to evaluate the performance.
>
> 2.Sorry for those confusing parts. In the data collection stage, we ask experts and doctors (not amateur crowd workers) to annotate all the corpus. Each annotator is paid from 0.5 to 4 RMB per instance. In the human evaluation in section 4.3, we ask the amateur workers to evaluate human performance. Moreover, all workers are trained before annotating instances from the testing data.
>
> 3.Thanks for your advice and this is an important comment. We have explained the cases detailedly in section 4.3, especially the Chinese language characteristics.
>
> We will carefully revise all the unclear points in the paper.
>
> The “meaningless text” is found based on keyword matching and regular expression, such as the sentence "The criteria is as follows" is filtered. We also ask workers to re-check sampled text in CHIP_CTC.
>
> PII:  Sorry for the missing part. We have added the "PII and IRB" section for each dataset in the supplementary part.
>
> Diagnosis item sampled: The diagnosis items of CHIP-CDN are randomly sampled from the items not covered by the common medical synonyms dict. We have added the clarification in our paper.
>
> Documentation:
>
> 1. The textbooks are public publications. The dataset is constructed from the freely accessed electronic version, and the dataset is only for non-commercial use purposes.
> 2. We have contacted each data provider for the annotater and annotation expense/duration information, which are added in the supplementary part.
> 3. Sorry for the missing parts. We have revised the paper and add the details for questions (4d,5a/b/c) in the supplementary part. For 5.b, the option has been corrected to "N/A" since the datasets included in CBLUE are either from public resources or not refer to the ethics, which has been checked by the IRB committee of each data provider.
> 4. We have revised the paper and add the annotation process in the supplementary part.
>
> We hope to construct the benchmark with diverse data sources that can better evaluate the model biomedical language understanding; thus, we include the KUAKE dataset, consisting of real-world medical search logs.

---

### Author Response · Authors · 2021-07-14
**Response to All Reviewers**

We would like to express our great appreciation to reviewers for their comments on our paper. In this paper, we propose a Chinese biomedical NLU benchmark with baselines and in-depth documentation. Sorry for some confusing parts in the previous version of the paper. We would like to highlight some details in the data collection and human evaluation procedure.

In the data collection stage, experts, doctors (not amateur crowd workers), and students with medical backgrounds (from medical college) are employed to annotate all the corpus. Each annotator is paid from 0.5 to 4.0 RMB per instance according to the task complexity. We maintain strict and high criteria for approval and review at least 10 random samples from each worker to decide whether to approve or reject.

In the human evaluation in section 4.3, we ask the amateur workers to evaluate human performance. Moreover, all workers are asked to annotate some data from the development set; then, their annotations are validated against the gold standard. Annotators need to correct their annotation mistakes repeatedly so that they can master the specific tasks. Finally, they annotate instances from the testing data, and these annotations are used to compute the final human scores.

We think that the quality of the CBLUE benchmark is of great importance, and noisy instances may lead to a misrepresentation of a model's performance. So we take great effort to do the annotation job to ensure data quality.

---

### Decision · Program_Chairs · 2021-07-26

**Decision:**

Reject

**Comment:**

The reviewers feel that the new dataset is interesting but the contributions are not sufficient for acceptance. (1) The authors need to clarify which part was annotated by trained professionals and which part was annotated by amateurs. (2) The authors need to compare CBLUE and CLUE.